

# A step toward building a unified framework for managing AI bias

Saadia Afzal Rana[1], Zati Hakim Azizul[1] and Ali Afzal Awan[2]

[1] Department of Artificial Intelligence, Faculty of Computer Science and Information Technology, Universiti Malaya, Kuala Lumpur, Malaysia
[2] National University of Science and Technology, Islamabad, Pakistan

## ABSTRACT

Integrating artificial intelligence (AI) has transformed living standards. However, AI's efforts are being thwarted by concerns about the rise of biases and unfairness. The problem advocates strongly for a strategy for tackling potential biases. This article thoroughly evaluates existing knowledge to enhance fairness management, which will serve as a foundation for creating a unified framework to address any bias and its subsequent mitigation method throughout the AI development pipeline. We map the software development life cycle (SDLC), machine learning life cycle (MLLC) and cross industry standard process for data mining (CRISP-DM) together to have a general understanding of how phases in these development processes are related to each other. The map should benefit researchers from multiple technical backgrounds. Biases are categorised into three distinct classes; pre-existing, technical and emergent bias, and subsequently, three mitigation strategies; conceptual, empirical and technical, along with fairness management approaches; fairness sampling, learning and certification. The recommended practices for debias and overcoming challenges encountered further set directions for successfully establishing a unified framework.

# INTRODUCTION

Data-driven decision-making applications have been deployed in vital areas such as finance (*Baum, 2017*), judiciary (*Angwin et al., 2016*; *Abebe et al., 2020*), employment (*Zhao et al., 2018b*), e-commerce (*Weith & Matt, 2022*), education (*Kim, Lee & Cho, 2022*), military intelligence (*Maathuis, 2022*) and health (*Gardner et al., 2022*). On one side, the potential of AI is widely recognised and appreciated. On the other side, there is significant uncertainty in managing negative consequences and subsequent challenges (*National Institute of Standards and Technology, 2022*), and a major hindrance to progress is a bias enrooted throughout the AI pipeline's development process (*Fahse, Huber & Giffen, 2021*).

The consequences of unwanted discriminatory and unfair behaviours can be detrimental (*Pedreshi, Ruggieri & Turini, 2008*), adversely affecting human rights (*Mehrabi et al., 2021*), university admissions (*Bickel, Hammel & O'Connell, 1975*), profit and revenue (*Mikians et al., 2012*) and facing legal risks (*Pedreshi, Ruggieri & Turini, 2008*; *Northeastern Global News, 2020*; *Romei & Ruggieri, 2014*). "Bias has existed since the dawn of society" (*Ntoutsi et al., 2020*). However, AI-based decision-making is criticised for introducing different types of biases. The ever-growing worries demand AI-based systems

Corresponding author
Zati Hakim Azizul, zati@um.edu.my

to rebuild technical strategies to integrate fairness as a core component of its infrastructure. Fairness is the absence of bias or discrimination. An algorithm that makes biased decisions against specific people is considered unfair (*Fenwick & Molnar, 2022*). The ethical implications of systems that affect individuals' lives have sparked concerns regarding the need for fair and impartial decision-making (*Pethig & Kroenung, 2022*; *Hildebrandt, 2021*). Consequently, extensive research has been conducted to address issues of bias and unfairness, while also considering the limitations imposed by corporate policies, legal frameworks (*Zuiderveen Borgesius, 2018*), societal norms, and ethical responsibilities (*Hobson et al., 2021*; *Gupta, Parra & Dennehy, 2021*).

There is still a lack of an organised strategy for tackling potential biases (*Richardson et al., 2021*). When looking for the origin of bias in AI decision-making, it is prevalent that the issue is either data or algorithms, and the root cause is humans. Humans transmit cognitive bias while creating/generating data or designing algorithms (*National Institute of Standards and Technology, 2022*; *Fenwick & Molnar, 2022*). Thoroughly evaluating existing literature is partial to this work in learning fairness management towards proposing a unified framework to address prejudice and its subsequent mitigation method.

In order to gain a comprehensive understanding, we frame our research by mapping the software development life cycle (SDLC), machine learning life cycle (MLLC) and cross industry standard process for data mining (CRISP-DM) process model to have a general understanding of how phases in this development process are related to each other. We categorise bias into three classes; pre-existing, technical and emergent bias and, subsequently, three mitigation strategies; conceptual, empirical and technical, along with fairness management approaches; fairness sampling, learning and certification. We discuss them in light of their occurrence at a specific phase of the development cycle. The recommended practices to avoid/mitigate biases and the challenges encountered in the course of action to address them are discussed in the later part.

## RATIONALE

A foundation must be established to accommodate all perplexing features, and it is necessary to study various facets of bias and fairness from the perspective of software engineering integration in AI. This study maps together the software development life cycle (SDLC), machine learning life cycle (MLLC), and cross industry standard process for data mining (CRISP-DM) processes. The mapping provides a general understanding of how phases in these software engineering (SE) and AI development processes relate and can benefit from SE's best practices within AI. The proposed framework aims to detect, identify, and localize biases on the spot and prevent them in the future by comprehending their core causes. The framework handles bias as a defect management process in software engineering.

## THE AUDIENCE IT IS INTENDED FOR

We believe we are the first to present this innovative framework. The framework can help ML researchers, ML engineers, data analysts, data scientists, software engineers, and software architects develop superior versions of software applications with higher

accuracy, better defect tracking, swifter control test timings and faster time-to-market release.

## SURVEY/SEARCH METHODOLOGY SECTION

Integrating bottom-up and top-down research methodologies was used to gather publications on bias in AI/ML applications. Each co-author gathered pertinent material and added it to a shared repository. The keywords for search are optimised in each search with an expectation to shortlist articles containing exact information. The three core domains of artificial intelligence, machine learning, and software engineering were the focus of the search, as shown in Fig. 1. Furthermore, 'AND' and 'OR' string operators were used along with double quotation marks to further narrow down to search fairness in AI/ML.

Interestingly, each survey introduced some new types of bias or fairness definitions. Only articles that detail bias in data, algorithms, assessment tools, fairness management approaches, bias detection, identification, mitigation strategy, fairness matrices, AI/ML/SE development cycles, datasets characteristics, AI ethics and principles issues were shortlisted during the inclusion criteria. We are further guided to our goal by sections from academic books, keynote addresses by renowned speakers (some of them are J. Buolamwini, T. Gebru (*Buolamwini & Gebru, 2018*) and C. O'Neil (*O'Neil, 2017*)), and various advisories published. Non-tabular data, domain/language-dependent technology, and the absence of experimental results are among the exclusion factors that reduce the number of articles we can find from 254 to 72. Due to a lack of mapping and tackling a variety of interlinked biases, we faced many challenges, *i.e.*, uncoordinated development efforts, inefficient use of resources, poor quality of ML models, limited scalability, amplification of biases, unfair decision-making, a lack of diversity, a loss of trust, and ethical issues. And in short found difficulties in integrating ML models into software. Addressing these issues requires a multifaceted approach that involves addressing biases at multiple levels, including in the data, the algorithms and the decision-making processes. It also requires a commitment to ethical and responsible development and use of technologies, especially when all development cycles should be synchronised together.

## REVIEW OF LITERATURE

The literature review on the subject is dense, with many forms of biases and mitigation strategies, most of which are interlinked. It is essential to be aware of potential biases in the AI pipeline and the appropriate mitigation strategies to address undesirable effects (*Bailey et al., 2019*).

### Realistic case study of AI bias

The previous 10 years have seen AI's widespread adoption and popularity, allowing it to permeate practically every aspect of our daily lives. However, safety and fairness concerns have prompted practitioners to prioritise them while designing and engineering AI-based applications (*Chouldechova et al., 2018*; *Howard & Borenstein, 2018*; *Osoba & Welser, 2017*). Researchers have enumerated a few applications that, because of biases, have a

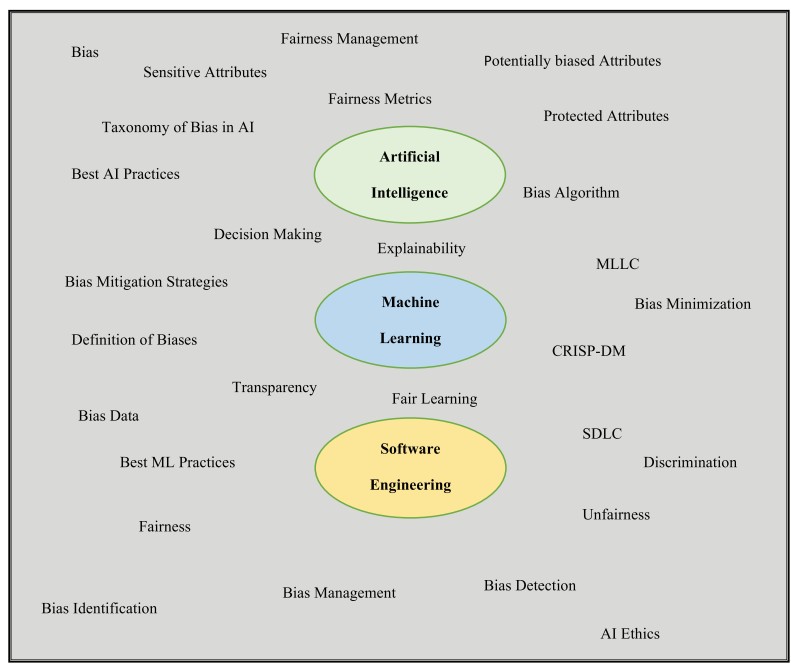

**Figure 1 Search words across three predefined domains-AI, ML & SE.**

detrimental impact on people's lives, such as biometrics apps, autonomous vehicles, AI chat-bots, robotic systems, employment matching, medical aid, and systems for children's welfare.

The US judiciary uses the correctional offender management profiling for alternative sanctions (COMPAS) (*Pagano et al., 2023*), an AI-based tool, to identify offenders more likely to commit crimes again. However, the Pulitzer Prize-winning nonprofit news organisation ProPublica (*Angwin et al., 2016*) discovered that COMPAS was racially prejudiced and that black criminals were at high risk (*Dressel & Farid, 2018*). PredPol, or predictive policing, is an AI-enabled software to predict the next crime location based on the number of arrests and frequency of calls to the police regarding different offences. PredPol was criticised for its prejudiced behaviour as it targeted racial minorities (*Benbouzid, 2019*).

An AI algorithm known as the Amazon recruiting engine was developed to evaluate the resumes of job applicants applying to Amazon and then shortlist eligible candidates for further interviews and consideration. However, it turned out that the Amazon algorithm was discriminatory regarding hiring women (*Hofeditz et al., 2022*). A labelling tool in Google Photos adds a label to a photo that corresponds to whatever is seen in the image. When it referred to images of a black software developer and his friend as gorillas, it was determined to be racist (*González Esteban & Calvo, 2022*). The well-known programme StyleGAN (*Karras, Laine & Aila, 2019*) automatically creates eerily realistic human faces and produces white faces more frequently than racial minorities.

Biases based on racial, gender, and other demographic factors restrict communities from using AI-automated technologies for regular tasks in the health and education sectors

(*Egan et al., 1996*; *Brusseau, 2022*). Organisations must know the different types of biases in their data/algorithm that can affect their machine learning (ML) models. Ultimately, identifying and mitigating biases that skewed or produced undesirable outcomes and impeded the advancements made by AI for the everyday person is more than necessary.

## Assessment tools

A systematic effort has been made to provide appropriate tools for practitioners to adopt cutting-edge fairness strategies into their AI pipelines. Software toolkits and checklists are the two basic ways to ensure fairness (*Richardson & Gilbert, 2021*). Programming language functions that can be used to identify or lessen biases are known as toolkits. AI practitioners can employ checklists and detailed instructions by fairness specialists to ensure ethical consideration is incorporated across their pipelines (*Bailey et al., 2019*).

Fairness indicators and the What-If toolkit (WIT) are well-known tools that Google provides (*Richardson et al., 2021*). Fairness indicators are based on fairness metrics for binary and multiclass classifiers (*Agarwal, Agarwal & Agarwal, 2022*). The What-If tool is an interactive visual tool designed to examine, evaluate, and compare ML models (*Richardson et al., 2021*). Uchicago's Aequitas is utilised to assess ML-based outcomes to identify various biases and make justified choices regarding the creation and implementation of such systems (*Saleiro et al., 2018*). IBM's AI Fairness 360 is a toolkit providing fairness detection and mitigation strategies (*Bellamy et al., 2019*). LinkedIn's fairness toolkit (LiFT) provides detection strategies for measuring fairness across various metrics (*Brusseau, 2022*). Microsoft's Fairlearn, ML Fairness Gym, Scikit's fairness tool, and PyMetrics Audit-AI are readily available tools to detect/mitigate (or both) bias and ensure fairness management. These assessment techniques expand practitioners' options for developing ethical products and maintaining stakeholder confidence (*Dankwa-Mullan & Weeraratne, 2022*).

Despite the availability of tools that incorporate explainable machine learning methods and fair algorithms, none of them currently offers a comprehensive set of guidelines to assist users in effectively addressing the diverse fairness concerns that arise at different stages of the machine learning decision pipeline (*Pagano et al., 2023*). Moreover the responsibility for identifying and mitigating bias and unfairness is often entirely placed on the developer, who may not possess sufficient expertise in addressing these challenges and cannot be solely responsible for the same (*Schwartz et al., 2022*). There is lack of consensus on what constitutes fairness as there is no single definition of fairness. Moreover, these tools have limited ability to detect complex forms of bias as they are designed to detect bias due to protected attributes only and casual fairness demands to investigate underlying relations between different attributes. Fairness assessment tools can themselves be biased. For example, a tool that is designed to detect bias in natural language processing models may be biased towards certain languages or dialects. Sometimes they can be time-consuming and expensive to use for organizations to adopt these tools, especially small or resource-constrained organizations. One more notable objection regarding them is that it can be difficult to interpret the results which make it difficult for organization to take

action to address any bias that is found. Majority of fairness assessment tools may not be able to identify the root cause of bias.

### Fairness management research datasets

Several standard datasets are available to make research on bias and fairness easier. Each of them has sensitive or protected attributes that could be used to demonstrate unfair or biased treatment toward underprivileged groups or classes. Table 1 summarises some of these datasets' characteristics. These datasets are used as a benchmark to evaluate and compare the effectiveness of bias detection and mitigation strategies.

## BROAD SPECTRUM OF BIAS

Since bias has existed for as long as human civilisation, research on the topic is popular. The literature review, however, is full of various terminologies and theories that either overlap or are interlinked, and this conflict is enough to perplex researchers. In this section, we will try to throw light on a different aspect of bias in AI with a perspective to overcome confusion and increase researchers' understanding. This effort sets the groundwork for building a unified framework for fairness management.

### Bias, discrimination and unfairness

The existence of bias, discrimination, and unfairness are related topics, but it is essential to distinguish them for further investigation/analysis (*Future Learn, 2013*). "Bias is the unfair inclination or prejudice in a judgement made by an AI system, either in favour of or against an individual or group" (*Fenwick & Molnar, 2022*). Discrimination can be considered a source of unfairness due to human prejudice and stereotyping based on sensitive attributes, which may happen intentionally or unintentionally. In contrast, bias can be considered a source of unfairness due to the data collection (*Luengo-Oroz et al., 2021*), sampling, and measurement (*Li & Chignell, 2022*). Discrimination is a difference in the treatment of individuals based on their membership in a group (*Ethics of AI, 2020*). Bias is a systematic difference in treating particular objects, people or groups compared to others. Unfairness is the presence of bias where we believe there should be no systematic difference (*Zhao et al., 2018b*). Bias is a statistical property, whereas fairness is generally an ethical issue.

### Protected, sensitive and potentially biased attributes

Until we stop making it, machine bias will keep appearing everywhere: bias in, bias out (*Alelyani, 2021*). Protected attributes are characteristics that cannot be relied upon to make decisions and may be chosen based on the organisation's objectives or regulatory requirements. Similarly, sensitive attributes are characteristics of humans that may be given special consideration for social, ethical, legal or personal reasons (*Machine Learning Glossary: Fairness, (n.d.)*). In literature, these terms are used interchangeably and in place of one another. It is a potentially biased attribute if changing the value of an attribute through the alternation function has an impact on prediction, such as altering a male attribute to a female one or a black attribute to a white one (*Alelyani, 2021*). Any one of these attributes in the dataset necessitates special consideration. Sex, race, color, age,

**Table 1 Few popular datasets, along with their characteristics.**

| Dataset name | Attributes | No. of records | Area |
|---|---|---|---|
| COMPAS (*Bellamy et al., 2019*) | Criminal histories, jail & prison times, demographics, COMPAS risk scores | 18,610 | Social |
| German credit (*Pagano et al., 2023*) | Housing status, personal status, amount, credit score, credit, sex | 1,000 | Financial |
| UCI adult (*Schwartz et al., 2022*) | Age, race, hours-per-week, marital status, occupation, education, sex, native country | 58,842 | Social |
| Diversity in faces (*National Institute of Standards and Technology, 2022*) | Age, pose, facial symmetry and contrast, craniofacial distances, gender, skin color, resolution along with diverse areas and ratios | 1 million | Facial images |
| Communities and crime (*Dua & Graff, 2017*) | Crime & socio-economic data | 1,994 | Social |
| Winobias (*Rhue & Clark, 2020*) | Male or female stereotypical occupations | 3,160 | Coreference resolution |
| Recidivism in Juvenile justice (*Merler et al., 2019*) | Juvenile offenders' data and prison sentences | 4,753 | Social |
| Pilot parliaments benchmark (*Redmond, 2011*) | National parliaments data (*e.g.*, gender and race) | 1,270 | Facial images |

marital status, family, religion, sexual orientation, political opinion, pregnancy, physical or mental disability, career responsibilities, social origins, and national extraction are a few attributes examples (*Fair Work Ombudsman, (n.d.)*). However, it has been observed that groups/individuals still face discrimination through proxy attributes (*Chen et al., 2019*), even in the absence of some protected or sensitive traits. Proxy attributes correlate with protected or sensitive attributes, such as zip code (linked with race) (*Mehrabi et al., 2021*).

## Trio-bias feedback loop

Data is the primary driving force behind most AI systems and algorithms, so they need data to be trained. As a result, the functioning of these algorithms and systems is closely tied to data availability. Underlying training data bias will manifest through an algorithm's predictions (trained on it). Furthermore, algorithms may reflect discriminatory behaviour based on specific design considerations even when the data is fair. The biased algorithm's output can then be incorporated into the existing system and impact users' choices, producing more biased data that can be used to train the new algorithm. Figure 2 depicts the feedback loop between data biases, algorithmic biases, and user involvement. Humans are involved in data preparation and algorithm design; therefore, whether bias results from data or an algorithm, humans are the root cause.

## Categorisation of biases

The human bias has more than 180 varieties, for example *Dabas (2021)*. Bias has been broadly divided into three major categories to accomplish fairness with a focus on detection and mitigation mechanisms and dispel confusion with many different types of bias (*Gan & Moussawi, 2022*). Bias categories include pre-existing, technical and emergent (*Friedman et al., 2013*).

A bias before the development of technology is referred to as pre-existing bias. This kind of bias has its roots in social structures and manifests itself in individual biases. The same is introduced into technology by people and organisations responsible for its development (*Curto et al., 2022*), whether explicitly or implicitly, consciously or unconsciously (*Friedman et al., 2013*). The AI literature identifies that the most typical biases are pre-

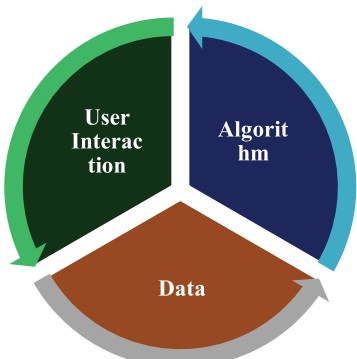

**Figure 2 Trio bias feedback loop among data, algorithm and user.**

existing (*Gan & Moussawi, 2022*). A few examples of pre-existing bias are the wrong model of Microsoft bot (*Victor, 2016*), Duane Buck's murder (*O'Neil, 2017*), the stance on fairness (*O'Neil, 2017*), criminal justice models (*Bughin et al., 2018*), and the understanding of concepts and reasonability of $CO_2$ emissions (*Luengo-Oroz, 2019*).

Technical bias refers to concerns about a product's technological design, such as technical limitations or decisions (*Friedman et al., 2013*). Technical issues faced in prominent software like IMPACT, Tech, LSI-R, Kyle's job application and hiring algorithms are because of technical bias. Emergent bias emerges after the practical use of a design as a result of a shift in social awareness or cultural norms (*Friedman et al., 2013*). PredPol, St. George's model, COMPAS, and facial analysis tech were adversely affected due to emergent bias.

### Origins of biases within the development cycle

The literature regarding the causes of biases and potential mitigation strategies is still scattered, and work on a systematic methodology for dealing with potential biases (*Nascimento et al., 2018*) and building well-established ML frameworks is in progress (*Suresh & Guttag, 2021*; *Ricardo, 2018*; *Silva & Kenney, 2019*). In this article, we attempt to overcome confusion and enhance understanding regarding the origin of different categories of bias during the development cycle by mapping SDLC, MLLC and CRISP-DM on a single scale, as shown in Fig. 3.

Professional designers or developers introduce pre-existing bias into the technical process, which is why it drags into the modelling phase of the CRISP-DM cycle (*Barton et al., 2019*). Technical bias, as opposed to pre-existing bias, results from how problems in the technical design are resolved. The design process and model evaluation contain many areas where technical bias can be identified (*Friedman et al., 2013*; *Ho & Beyan, 2020*). Pre-existing and technical biases occur prior to and within the technical development process. However, emergent bias appears during the actual use of the technical product after development (*Barton et al., 2019*). Emergent bias can be detected before deployment (during testing) and is often the most obvious category of bias (*Gan & Moussawi, 2022*). The same type of bias has several names in the literature. Therefore, biases were

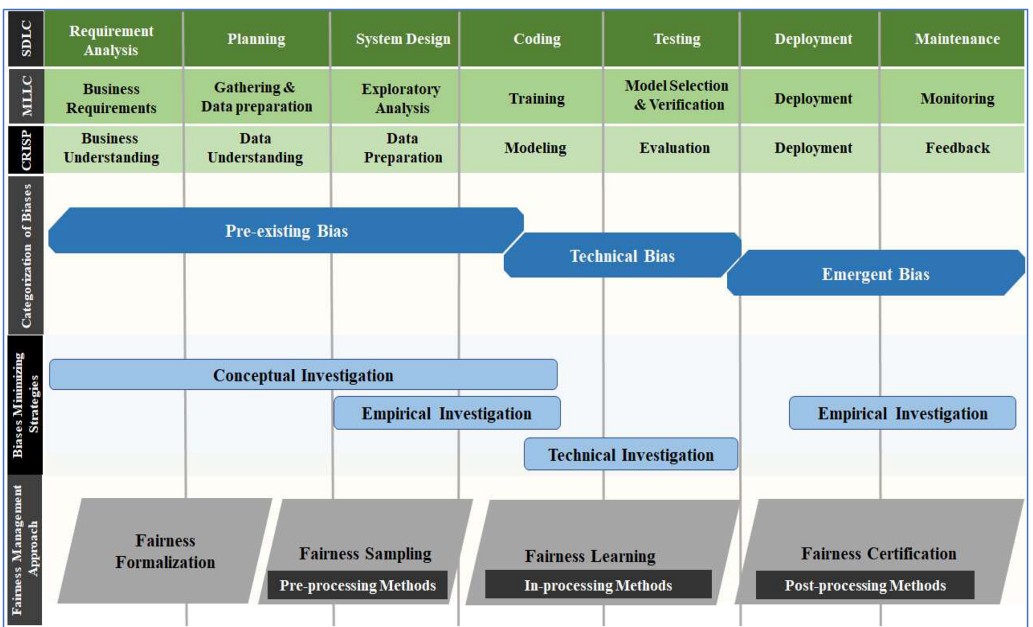

**Figure 3 Mapping the SDLC, MLLC and CRISP-DM across different biases categories, minimizing strategies and fairness management.**

descriptively synthesised and characterised based on their origins (*Ho & Beyan, 2020*; *Fink, 2019*). Based on a thorough understanding of the development cycles, commonly encountered distinct types of biases were assigned to the phases based on their origin, as shown in Fig. 4.

The bias type and subtype distribution across different phases will create an appropriate detection method (*Fahse, Huber & Giffen, 2021*). Equal opportunity, equalized odds, conditional demographic disparity, disparate impact, Euclidean distance, Mahalanobis distance, Manhattan distance, demographic disparity, and different tools and libraries are used for bias detection (*Garg & Sl, 2021*).

## Bias minimising strategies

Value sensitive design (VSD) is a methodology that can contribute to understanding and addressing issues of bias in AI systems (*Simon, Wong & Rieder, 2020*) and to promote transparency (*Cakir, 2020*) and ethical principles in AI systems (*Dexe et al., 2020*). It is a framework that provides recommendations for minimising or coping with various biases (*Gan & Moussawi, 2022*; *Friedman et al., 2013*), is adopted in this research as a potential strategy to minimise the biases associated with AI. In the VSD study, "value" is a general phrase that relates to what user values in life (*Atkinson, Bench-Capon & Bollegala, 2020*). This theoretically-based approach provides three possible solutions based on the types of investigations to prevent problems and advocate AI-specific value-oriented metrics that all stakeholders mutually agreed on.

Conceptual, empirical and technical are three different kinds of investigations. A conceptual investigation primarily focuses on analysing or prioritising different stakeholders' values in the design and use of technology (*Barton et al., 2019*; *Floridi, 2010*).

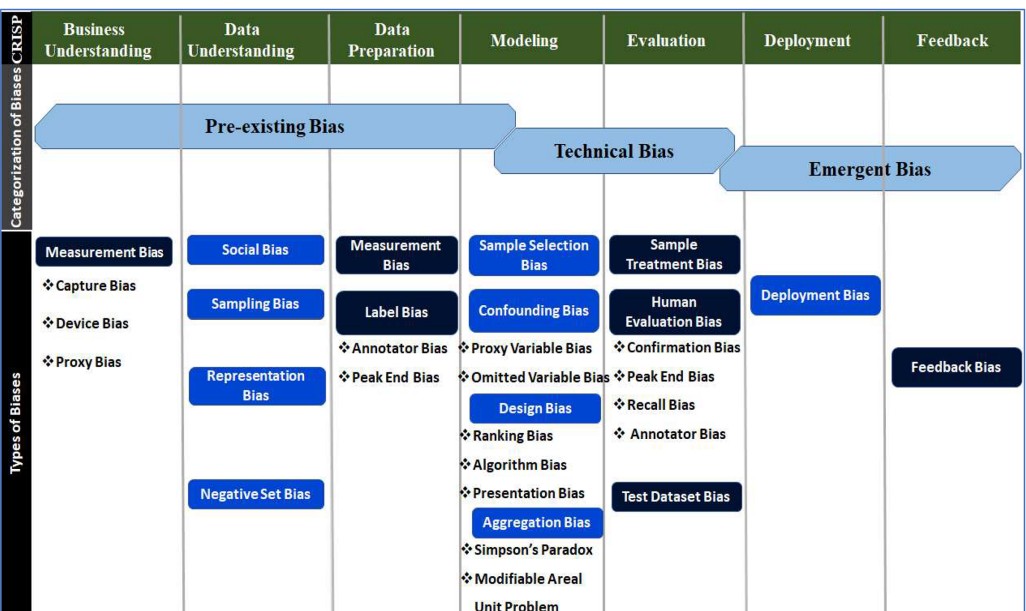

**Figure 4  Distribution of biases w.r.t categories across phases of the CRISP-DM development cycle.**

Empirical investigation assesses a technical design's effectiveness using factors like how people react to technological products (_Barton et al., 2019_; _Floridi, 2010_). It frequently entails observation and recording; quantitative and qualitative research methods are appropriate. Because of these characteristics, the phases are limited to data understanding (slightly), data preparation, modeling (to some extent), deployment and feedback.

The primary focus of the technical investigation is on technology while focusing on how technological characteristics and underlying mechanics promote or undermine human values and involves proactive system design to uphold values discovered in conceptual investigations (_Gan & Moussawi, 2022_; _Friedman et al., 2013_; _Ho & Beyan, 2020_). VSD has revealed the relationships between different types of investigations and types of AI biases (_Umbrello & Van de Poel, 2021_). Researchers concluded by recognising the value of conceptual and empirical investigation for addressing pre-existing bias, investigations for minimising technical bias, and technical and empirical investigation for tackling emerging bias (_Gan & Moussawi, 2022_).

## Bias mitigation methods

Several methods can mitigate a single bias, and multiple biases can be mitigated by a single method. The socio-technical approach comprising technical and nontechnical methods is widely adopted to counterattack the harmful effects of biased decisions (_Fahse, Huber & Giffen, 2021_). This approach mitigates bias and prevents it from recurring in the future. Table 2 depicts which method effectively mitigates different types of bias whenever it appears in different stages of the AI development cycle. The mitigation process is not executed during Feedback. From risk management's perspective, to evaluate an AI system's

**Table 2 A socio-technical approach for bias mitigation across the CRISP-DM development cycle.**

| | Business understanding | Data understanding | Data preparation | Modelling | Evaluation | Deployment |
|---|---|---|---|---|---|---|
| Measurement bias | Team diversity, exchange with domain expert | Proxy estimation | Rapid prototyping | | | |
| Social bias | | | Learning fair representation, rapid prototyping, reweighting, optimized preprocessing, data massaging, disparate impact remover | Adversarial debiasing, multiple models, latent variable model, model interpretability equalized odds, prejudice remover | | |
| Sampling bias | | | | Resampling | | Randomness |
| Representation bias | Team diversity | Data plotting, exchange with domain experts | Reweighing, data augmentation | Model interpretability | | |
| Negative bias | | | Cross dataset generalization | Bag of words | | |
| Label bias | | Exchange with domain experts | Data massaging | | | |
| Sample selection bias | | | Reweighing | | | |
| Confounding bias | | | | | | Randomness |
| Design Bias | | | Rapid prototyping | Exchange with domain experts, resampling, model interpretability, multitask learning | | |
| Sample treatment bias | | | | Resampling | Data augmentation | |
| Human evaluation bias | | | | Resampling | Representative benchmark subgroup validity, data augmentation | |
| Test dataset bias | | | Data augmentation | | | |
| Deployment bias | Team diversity, consequences in context | | Rapid prototyping | | | Monitoring plan, human supervision |
| Feedback bias | | | | | | Human supervision, randomness |

performance, 'Fairness' outclass any other vital measure, *i.e.*, dependability, efficiency, and accuracy (*Barton et al., 2019*; *Suri et al., 2022*).

Various strategies have been put forth by researchers and practitioners to address bias in AI. These strategies encompass data pre-processing, model selection, and post-processing decisions. However, each of these methods has its own set of limitations and difficulties, such as the scarcity of diverse and representative training data, the complexity of identifying and quantifying different forms of bias and the potential trade-offs between fairness and accuracy (*Pagano et al., 2023*). Additionally, ethical considerations arise when determining which types of bias to prioritize and which groups should be given priority in the mitigation process (*Pagano et al., 2022*). Bias mitigation methods can be complex, computationally expensive and can introduce new biases for example, a method that tries to balance the representation of different groups in a dataset may introduce a new bias in

favor of the majority group. They can be brittle such as that they can be sensitive to changes in the data or the model. This can make it difficult to ensure that the model remains fair over time. Another notable aspect that needs to be addressed are that there is a lack of understanding of the long-term effects of bias mitigation methods & standardized evaluation metrics. These methods can be opaque as it can be difficult to understand how they work, subsequently it can make it difficult to trust these methods, and to ensure that they are not introducing new biases. They might feel it difficult to adapt to new tasks, as result of it, they may not be effective for all machine learning models.

Despite these obstacles, the mitigation of bias in AI is of utmost importance to establish just and equitable systems that benefit everyone in society (*Balayn, Lofi & Houben, 2021*). Continuous research and development of mitigation techniques are crucial to overcome these challenges and ensure that AI systems are employed for the welfare of all individuals (*Huang et al., 2022*).

# FAIRNESS MANAGEMENT APPROACH

Bias and fairness are two mutually exclusive aspects of reality. In the absence of a unified definition, an absence of bias or preference for individuals or groups based on their characteristics is generally regarded as 'fairness'. It is necessary to do more than mitigate any bias detected to ensure that an AI system may be regarded as fair. Instead, "fairness-aware" system design should be encouraged (*Orphanou et al., 2021*). This approach incorporates "fairness" as a crucial design component from conception to deployment (sometimes extended to maintenance or upgradation). Figure 5 shows the fairness management approach and several implementation methods across the CRISP-DM model. The following steps constitute the fairness management approach:

## Fairness formalisation

Constraints, measures, specifications and criteria to ensure fairness is defined during the business understanding and data understanding phases (*Northeastern Global News, 2020*), as depicted in Fig. 5. In fact, at this stage, the benchmarks for auditing or evaluating fairness are stated.

## Fairness sampling

Fairness sampling generally refers to preprocessing skewed data through different methods, such as oversampling (*Orphanou et al., 2021*). Issues with data, *i.e.*, inaccuracy, incompleteness, improperly labelling, too much/less, inconsistency, and silos, are addressed at this stage (*Walch, 2020*). Fair sampling is accomplished during the data understanding and data preparation phases.

## Fairness learning

It is not always feasible to develop a fair model by eliminating the bias in the initial data before training. Designing a fair classifier that uses a fair algorithm is the solution in such scenarios (*Acharyya et al., 2020*). As a result, we can still use a biased dataset to train the model, and the fair algorithm still produces predictions through in-processing methods carried out during the modelling and evaluation phases (*Acharyya et al., 2020*).

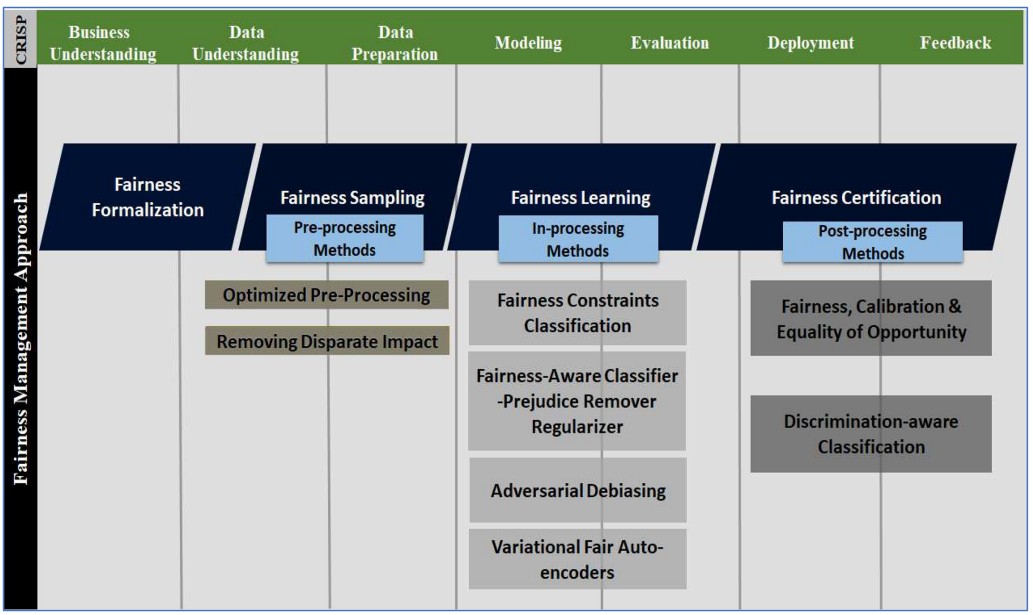

**Figure 5 Fairness management approach across multiple phases of the CRISP-DM development cycle.**

## Fairness certification

At the final stage of the testing and deployment phases, it is evaluated whether a prediction aligns with the criteria mentioned in the fairness formalisation phase by executing post-processing methods (*Floridi, 2010*). Fairness certification solutions verify that unfairness does not surface during the feedback phase.

## Limitations and challenges of existing approaches

After thoroughly examining various aspects of tools, methodologies, frameworks, and fairness solution spaces, it is now appropriate to consolidate and summarize the overall major limitations and challenges encountered throughout this exploration:

1. One of the shortcomings of current approaches is the lack of transparency and interpretability (*Srinivasu et al., 2022*). Many tools and strategies employed to mitigate biases in various domains, such as machine learning algorithms or content moderation systems, often lack clear explanations of how they address bias (*Feuerriegel, Dolata & Schwabe, 2020*). This lack of transparency makes it difficult for users to understand the underlying biases being addressed and the effectiveness of the applied methods (*Berente et al., 2019*). Additionally, without proper transparency (*Society to Improve Diagnosis in Medicine, 2021*), it becomes challenging to identify potential unintended consequences or biases that may arise from the bias management methods themselves (*Turney, 1995*).

2. Another major shortcoming lies in the limited customization and adaptability (*Basereh, Caputo & Brennan, 2021*). Most tools and strategies are developed with a one-size-fits-all approach, which may not adequately account for the specific biases

prevalent in different contexts or domains (*Qiang, Rhim & Moon, 2023*). This limits their effectiveness in managing biases that are nuanced and context-dependent. Furthermore, these methods often do not provide enough flexibility for users to customize and fine-tune the bias management mechanisms according to their specific needs and requirements.

3. Existing methods, tools, and strategies for bias management rely on manual intervention, which makes the process time-consuming and prone to human error. When bias management is carried out manually, it becomes difficult to ensure consistent and comprehensive coverage of all potential biases. Additionally, manual methods may lack scalability and efficiency, particularly when dealing with large datasets or complex models (*Chhillar & Aguilera, 2022*). Therefore, there is a need for more automated and robust approaches to bias management.

4. Bias can be unintentionally introduced when the dataset used to train AI models is not representative of the real-world population it is designed to serve (*Schwartz et al., 2022*). For example, if a facial recognition system is trained primarily on data from lighter-skinned individuals, it may exhibit higher error rates for darker-skinned individuals. This lack of diversity can perpetuate existing societal biases and lead to discriminatory outcomes (*Delgado et al., 2022*; *Michael et al., 2022*).

5. Existing methodologies tend to place a heavy emphasis on technical solutions for bias mitigation, often neglecting the importance of interdisciplinary collaboration (*Madaio et al., 2022*). Addressing bias in AI requires input from diverse stakeholders (*Michael et al., 2022*), including ethicists, social scientists, and policymakers. Without incorporating a multidisciplinary perspective, frameworks may overlook crucial ethical considerations and fail to account for the broader societal impact of AI systems.

6. Many existing strategies tend to oversimplify the concept of bias, reducing it to a binary problem. They often focus on mitigating only explicit biases while overlooking implicit biases, which are more subtle and deeply ingrained in societal structures (*Peters, 2022*). Addressing implicit biases requires a more nuanced understanding of the underlying social dynamics and power structures. Failure to consider these complexities can result in incomplete or ineffective bias mitigation strategies.

7. Identifying and mitigating the various interlinked biases that can arise in AI systems poses a significant challenge due to their diverse nature and complexity.

8. In some cases, there may be a potential trade-off between fairness and accuracy. For example, if an AI system is designed to be fair to all groups, it may not be as accurate as it could be.

9. There are ethical considerations around how to mitigate bias in AI systems (*Straw, 2020*). For example, should we prioritize fairness to individuals or to groups? Should we focus on mitigating historical bias or on preventing future bias.

10. The prevailing focus of most approaches lies in addressing bias reactively, leading to high costs associated with corrective measures. There is a pressing need to adopt a proactive stance and mitigate bias as soon as it is identified.

Lack of comprehensive evaluation frameworks is another hurdle in way to achieve fairness. While some methods may claim to address biases, there is often a lack of standardized evaluation frameworks to assess their effectiveness and potential trade-offs (*Landers & Behrend, 2023*). This absence of robust evaluation frameworks hinders the attainment of fairness on a global scale.

## RECOMMENDED PRACTICES TO AVOID/MITIGATE BIAS

After understanding the categorisation and minimising strategies of biases and knowing how to implement a fairness management approach during the model development life cycle, it is time to look at best practices to avoid/mitigate bias and ensure fairness. A few of them are as follows:

- Considering human customs in AI use and promoting all concerned stakeholders on the board *i.e.*, developers, users/general public, policy makers *etc.* to achieve an effective strategy (*Floridi, 2010*).
- Domain-specific knowledge must be incorporated to detect and mitigate bias (*Srinivasan & Chander, 2021*).

  ○ When collecting data, it is vital to have expertise in extracting the most valuable data variables (*Pospielov, 2022*).
  ○ Be conscious of the data's sensitive features, including proxy features, as determined by the application (*Vasudevan & Kenthapadi, 2020*).
  ○ Datasets should, to the greatest extent possible, represent the actual population being taken into account (*Celi et al., 2022*). Data selection by random sampling can perform effectively (*Pospielov, 2022*).
  ○ Preprocess the data to guarantee the maximum possible level of accuracy while minimising relations between results and sensitive attributes or to present data preserving privacy (*Floridi, 2010*).
  ○ For annotating the data, appropriate standards must be specified (*Zhang et al., 2022*).
  ○ Consider crucial elements such as the data type, problem, desired outcome, and data size while choosing the best model for the data set (*Pospielov, 2022*).

- The right model choice is one of the critical components of fairness management. Compared to linear models, which give exact weights for each feature being taken into account, deep models like decision trees can more easily conceal their biases (*Barba, 2021*).
- Bias detection should be incorporated as a necessary component of model evaluation and focus on model accuracy and precision (*Schmelzer, 2020*).
- Track the models' performance in use and continually evaluate it (*Schmelzer, 2020*).
- Encourage all stakeholders to report bias and management should take it positively (*Shestakova, 2021*).

- In order to prevent perpetuating inequity, AI systems must be responsible during the design, development, evaluation, implementation, and monitoring phases (*Stoyanovich, Howe & Jagadish, 2020*).
- Process and result transparency should be defined in a way that can be interpretable without in-depth knowledge of the algorithm (*Seymour, 2018*).

## CHALLENGES, OPPORTUNITIES AND FUTURE WORK

There are several obstacles to overcome before ethical AI applications and systems are successfully developed, free from bias, and wholly engineered along fairness lines. Planning to overcome these obstacles sets the direction of future research. A few serious challenges are:

- Despite numerous approaches to detect and mitigate bias/unfairness, no absolute results are yet available for the cutting edge to handle each type of biasness (*Ntoutsi et al., 2020*; *Pagano et al., 2023*).
- A mathematical definition cannot express all notions of fairness (*Ntoutsi et al., 2020*). From the ML perspective, literature is enriched with various definitions of fairness. One of the unsolved research issues is how to combine these definitions and propose a unique notion of fairness (*Le Quy et al., 2022*). Achieving it can evaluate AI systems in more unified and comparable manners. The ideals of fairness are incompatible with the operationalisation of the "from equality to equity" concept (*Fenwick & Molnar, 2022*), which necessitates approaching the issue from an operational war posture (*Belenguer, 2022*). Moreover, it demands integrating social and political knowledge in the primary process as critical elements (*Rajkomar et al., 2018*; *Elizabeth, 2017*).
- The literature review has a lot to say about the bias/fairness of the data and algorithm used by data-driven decision-making systems. However, not all areas have received the same degree of research community attention, *i.e.*, classification, clustering, word embedding, semantic role labelling, representation learning VAE, regression, PCA, named entity recognition, machine translation, language model, graph embedding, coreference resolution, and community detection (*Mehrabi et al., 2021*).
- In the context of fairness-aware ML, exploratory analysis of datasets is still not practiced widely (*Le Quy et al., 2022*). Real, synthetic and sequential decision-making datasets are not adequately exploited.
- The role of sensitive/protected attributes in measuring the performance of predictive models has been studied a lot. However, the role of proxy attributes in the same perspective demands more research work (*Le Quy et al., 2022*).
- Most research being done at the moment focuses on techniques that reduce bias in the underlying machine learning models through the algorithm's approach. Due to this, a research gap may be filled by a data-centered approach to the subject (*Rajkomar et al., 2018*).

- The oldest benchmark dataset was gathered 48 years ago from nations with active data protection laws. However, to meet the demands of the modern day, general data quality or collection regulations still need to be researched and developed (*Ntoutsi et al., 2020*).

**Establishing an agile approach to address bias in AI systems:**

A framework for managing bias in AI systems should be created to encourage fairness, accountability, and transparency throughout the AI system lifetime while integrating software engineering best practices, in light of the challenges listed above. Effectively reducing bias requires a multifaceted, adaptable, transparent, scalable, accessible, interdisciplinary, and iterative approach.

Agile methods can be employed to address bias in AI systems. By adopting an agile approach, AI developers can continuously monitor and mitigate bias throughout the development lifecycle and all stakeholders will be well informed with current situation (*Benjamins, Barbado & Sierra, 2019*). Framework based on agile approach can not only mitigate bias at the spot in proactive manners but also eradicate it from appearing in future by eliminating all interlinked biases as (*Landers & Behrend, 2023*; *Caldwell et al., 2022*). To design a framework for fair AI several working variables play a key role. Let us explore some of these variables in detail:-

**Data collection and preprocessing**

The first working variable to consider is the data used to train AI models. It is essential to ensure that the data collected is representative and diverse (*Zhao et al., 2018a*), without any inherent biases (*de Bruijn, Warnier & Janssen, 2022*). Biases can emerge if the data reflects historical prejudices or imbalances (*Clarke, 2019*). Careful preprocessing is necessary to identify and address these biases to prevent unfair outcomes. For example, if a facial recognition system is trained predominantly on a specific demographic, it may exhibit racial or gender biases.

**Algorithmic transparency and explainability**

To design a fair AI framework, it is crucial to consider the transparency and explainability of the algorithms used (*Umbrello & Van de Poel, 2021*). Black-box algorithms that provide no insight into their decision-making processes can pose challenges in identifying and rectifying biases (*Clarke, 2019*). By promoting algorithmic transparency, stakeholders can understand how decisions are made and detect any unfairness in the system. Explainable AI techniques, such as providing interpretable explanations for decisions, can also enhance fairness and accountability (*Toreini et al., 2020*).

**Evaluation metrics**

Establishing appropriate evaluation metrics is another essential working variable in designing a fair AI framework. The metrics used to assess the performance of AI systems should go beyond traditional accuracy measures (*Buolamwini & Gebru, 2018*) and incorporate fairness considerations (*Clarke, 2019*). For instance, metrics like disparate impact, equal opportunity, and predictive parity can help identify and mitigate biases across different demographic groups. Evaluating AI systems on these fairness metrics ensures that fairness is a fundamental goal rather than an afterthought.

### Regular audits and monitoring

A fair AI framework requires ongoing audits and monitoring to identify and rectify biases that may emerge over time. Regular assessments can help evaluate the fairness of AI algorithms and models in real-world scenarios (*Landers & Behrend, 2023*). It allows for continuous improvement and ensures that any biases are promptly addressed. Organizations should establish mechanisms to monitor the performance of AI systems, collect feedback from users, and engage in iterative improvements to enhance fairness (*Saas et al., 2022*). Moreover, encourage continuous learning within your team about bias, fairness, and ethical AI. Continuously stay informed about the most recent research and best practices in this domain and ensure ongoing updates.

### Stakeholder inclusivity

Involving diverse stakeholders in the design and deployment of AI systems is a critical working variable for fair AI. Including representatives from different communities, demographic groups, and experts from various fields can help identify potential biases and ensure fairness (*Clarke, 2019*). It is crucial to consider the perspectives and experiences of those who may be disproportionately affected by AI systems. Such inclusivity can lead to a more comprehensive understanding of biases and result in fairer outcomes (*Johansen, Pedersen & Johansen, 2021*).

### Ethical guidelines and governance

Ethical guidelines and governance play a vital role in shaping the design of a fair AI framework (*Hildebrandt, 2021*). Organizations should establish clear policies and guidelines that explicitly address fairness concerns. These guidelines should define what constitutes fairness and provide actionable steps to ensure it is upheld. Integrating ethical considerations into the design and decision-making processes can help prevent biases (*Jobin, Ienca & Vayena, 2019*). In nutshell, agile methods provide a framework for effectively addressing bias in AI systems. By adopting diverse teams, defining clear goals and metrics, conducting frequent reviews and iterations, ensuring transparency, curating unbiased datasets, monitoring and evaluating system performance, collaborating with stakeholders, and prioritizing ethics, organizations can develop and deploy AI systems that are fair, unbiased, and inclusive.

## CONCLUSION

The presence of bias in our world is reflected in the data. It can appear at any phase of the AI model development cycle. It is not only the developers' core responsibility to ensure model fairness but AI fairness requires a collaborative effort from all stakeholders *i.e.*, policymakers, regulators (*Yavuz, 2019*), users, general public *etc.* for responsible and ethical AI solutions (*Rajkomar et al., 2018*). The future will see AI play an even more significant impact in both our personal and professional lives. Recognising the benefits and challenges of developing ethical and efficient AI models is crucial (*John-Mathews, Cardon & Balagué, 2022*). AI developers bring with them a variety of disciplines and professional experiences.

Narrowing the subject to a single profession or area of expertise would oversimplify the situation. Mapping SDLC, MLLC, and CRISP-DM on a single reference will boast an understanding of practitioners from various technical frameworks to a single point. Once the bias is identified and mitigated at each phase of the development process, the technical team will remain vigilant and unable to de-track or display negligence. The fairness management approach will further enhance the effectiveness of revealing the hidden shortcomings during the development process.

From this perspective, a firm grasp of standard practices will be the foundation for a unified AI framework for fairness management. Through the proposed framework, organisations of all sizes can manage the risk of bias (*Straw, 2020*) throughout a system's lifecycle and ensure that AI is accountable by design. In order to manage the associated risks with AI bias, the proposed framework ought to have the following key characteristics:

- Outlines a technique for conducting impact assessments
- Promotes better awareness of already-existing standards, guidelines, recommendations, best practices, methodologies, and tools and indicates the need for more effective resources.
- Integrate best practices from software engineering similar to 'defect management' to tackle bias as 'defect'. Then, detect, identify and localise bias on the spot before proceeding further and eliminating its chance of appearing in future.
- Ensure each stakeholder comprehends his or her responsibility.
- Sets out corporate governance structures, processes and safeguards that are needed to achieve desired goals.
- Law and regulation agnostic.
- In light of the field of AI's rapid growth, the framework should be updated to ensure it is up-to-date and adapted.

Briefly, AI fairness management is centered on a governance framework that encourages the prevention of bias from manifesting in a way that unjustifiably leads to less favorable or harmful outcomes and enables businesses to create more accurate and practical applications and more persuasive to customers. Overall, a framework for fair AI should provide a structured approach to manage biases and ensure that AI systems operate ethically, fairly, and transparently, while accommodating the complexities and challenges inherent in AI development and deployment.

### Funding

This work was supported by the Higher Education Commission Pakistan. The funders had no role in study design, data collection and analysis, decision to publish, or preparation of the manuscript.

## Grant Disclosures

The following grant information was disclosed by the authors:
Higher Education Commission Pakistan.

## Competing Interests

The authors have no conflicts of interest to declare. All co-authors have seen and agree with the contents of the manuscript and there is no financial interest to report.

## Author Contributions

- Saadia Afzal Rana conceived and designed the experiments, performed the experiments, analyzed the data, prepared figures and/or tables, authored or reviewed drafts of the article, and approved the final draft.
- Zati Hakim Azizul conceived and designed the experiments, performed the experiments, analyzed the data, prepared figures and/or tables, authored or reviewed drafts of the article, and approved the final draft.
- Ali Afzal Awan conceived and designed the experiments, performed the experiments, analyzed the data, prepared figures and/or tables, authored or reviewed drafts of the article, and approved the final draft.

## Data Availability

This is a literature review.

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
