# Peer review of "A step toward building a unified framework for managing AI bias"

_PeerJ Computer Science, doi:10.7717/peerj-cs.1630_

## Round 0.1 · original submission · Major Revisions

Both Reviewers are generally happy with the manuscript, but want several things fixed in the manuscript. Most of them have to do with a more detailed analysis (mostly highlighted by Reviewer 2) and further clarifications (mostly highlighted by Reviewer 1).

Reviewer 1 ·

Basic reporting

The use of English is good and the paper is easy to read. However, the language is too colorful for a scientific text. E.g., illustrious speakers (like who?) line 90, line 97 will not just perplex.., line 192 the literature is stuffed with, line 237 creep into...
This is not a pop science talk, but rather a scientific article that should make use of a neutral tone.

The value of some of the claims is questionable wrt supporting the content of the paper? E.g. lines 92-94, surely the IT field is very wide, and 15 years while a considerable time might be (or not) relevant per se to what is discussed in this paper. Authors should provide clear argumentation as to why this experience is relevant, what they have observed, is it anecdotal, if not can they point the reader to evidence that indeed supports the fact that this experience is relevant in the context of the paper.

I would also challenge the first sentence of the abstract and first claim of this paper. Has it really improved every aspect of our experiences? AI is a tool, and as such it can be used for good, or bad, or it can be neutral. There are countless examples on how it is making our lives worse, e.g., social media addiction (especially among youth), surveillance, tracking and manipulating people in the best case, and worst case tracking and infringing on the rights of employees.

Sentences in lines 120-122 and 123-124 basically say the same thing.

Experimental design

To what degree is the goal of the paper achieved (as stated in lines 50-52)?

Line 168, to say that we should making bias is analogous to me to asking programmers to code w/o bugs. So it is not very realistic, instead what we can do is first be aware in which ways we can be biased, and thereafter mitigate it.

Line 188-190. I think the way authors put this is too simplistic. E.g., if we have data the represents a certain state of affairs in a poor area, and we find correlations say between crime rates and ethnicity, that per se is not the fault of the human. The salient point here is what we do with these findings, are people working the data making generalizations, as well as mistaking correlation for causation, and thus making unfair decisions towards specific groups? Do the people working with the data care to look beyond the numbers? I.e., could we even out the disparities by putting more effort (money and other resources) in the poor areas, to give people living there a decent shot at life? And the fact that there is a problem to begin with is made visible by the data being as it is. The data shows how the state of affairs wrt some metrics is in the particular domain where it is recorded. Then our job is to properly handle this data, instead of using it to discriminate people already in vulnerable positions, it can be used to point our attention to areas of society that require our attention to improve. Then again, once the data is gathered we cannot guarantee that it will not be misused. And so the question becomes how can we evaluate the advantages and risks of gathering it in the first place?

Line 344. This again is very simplistic, as there are other actors interacting with the AI, like the intended or unintended users, that can interact with it in more subtle and sophisticated ways, that can lead to misuse. Developers have their responsibilities, as well the design processes themselves, but they are not the only relevant actors in this discussion.

Line 287 It seems that authors fall back to bias mitigation, as opposed to what it is claimed in 261 regarding fairness-aware design.

Validity of the findings

Authors do not provide much argumentation to justify why they have done the presented mappings in the way they have. The reader is left with looking at the figures, and connecting the dots themselves.
This makes it hard to properly evaluate the contribution.
The text here is more a listing of concepts discussed in literature.

The discussion lacks evaluation, even at the theoretical level, why is the proposal adequate (lines 354-358)? How do authors plan to actually evaluate the feasibility of this in practice?

·

Basic reporting

• The manuscript shows a clear, unambiguous, and professional use of the English language, making it easy to read and understand. However, there is no smooth transition between the ideas, or between the sections that form the structure of the manuscript.
• The authors provide proper context through a short Introduction and other sections that are related to the methodological approach. Although, the authors, while conducting the reader throughout their reasoning -and this is something they do along the entire corpus of the manuscript- rely entirely on the references they use. Consequently, many ideas are limited to a descriptive character instead of actually expressing a thought, in most cases, the sentences act as an advertisement of the referenced work.
• The manuscript showcases relevant and popular works regarding the examples they use for illustrating different uses and areas in which Data-Driven technologies are currently deployed. However, after reading the sequence of affirmations the authors use to present their thesis and reasoning, one can notice the absence of sources and an analysis to defend/contrast their points of view. By doing so, the authors introduced confirmation bias in their study, referencing -mainly- works that were in tune with their discourse or served it. Also, the work definitions presented for fairness, bias, etc. are provided not as a result of analysis but as an affirmation, resulting in simplistic and unitarian, orphan of any link with other variables present within the complex ecosystem of non-discrimination, fairness, bias, and trustworthiness in the context of automatic/algorithmic decision-making software solutions.
• The structure of the manuscript conforms to PeerJ standards.
• The manuscript presents a review that, as per is presented, does not need broad or cross-disciplinary arbitration, and adequately fits within the scope of the journal.
• The topic of algorithmic fairness in the context of AI is a field that has been reviewed recently from a multiplicity of angles and uses a wide range of approaches. The reviewed manuscript is original in mapping the correspondence of their opinions into the Machine Learning Life Cycle, and other more general software development methodologies. This is particularly helpful for software engineers. I would suggest analyzing, in more depth, IA and ML development specific methodologies, including those with an agile approach, as the manuscript explores a cycle based on a more traditional approach.

Experimental design

• The manuscript content is within the Aims and Scope of the journal.
• An investigation was performed to some level of technical standard.
• The methods were described with sufficient detail.
• The manuscript exhibits a partial survey of the subject, limited mainly by the authors’ confirmation biases in their approach to the topic. Is missing further analysis on the following elements:
o Working definitions for Bias, Fairness, and other variables present in the manuscript, in the context of AI and ML.
o The social/ethical impact of the working definitions previously referred, to on the AI and ML solutions ‘target populations. The results of such analysis are relevant when determining at which stage of the development cycle allocate which task.
o The different methodologies used to develop AI and ML solutions, and as a result, more accurate maps of the life cycle of these types of development projects.
• The reference sources were adequately cited, but poorly analyzed.
• The review was organized logically into coherent paragraphs, although these paragraphs that embodied the manuscript, and its subsections are not smoothly linked to each other.

Validity of the findings

The goals set as part of the introductory part of the manuscript are met.
There are no unsolved questions, gaps, or future research ventures identified as a result of the study.

---

## Round 0.2 · Major Revisions

Dear authors,

I agree with the reviewer's comments that in order to show case the contribution of your paper, your analysis/critique of existing literature on the topic needs to be included. Please consider carefully the comments given to you, and act upon them in your revised version.
Regards,

·

Basic reporting

The authors performed major aesthetic changes to the manuscript in correspondence to my previous comments regarding the organization of their ideas and paragraph structure. However, there are still unsolved issues:
Firstly, the manuscript is still a recollection of descriptive analysis about the references cited. It still lacks of the author’s clear vision about those cited related works. In their rebuttal letter, the authors claim the scope of their work is “…to draw the attention to possible interlinked biases…”, and also that “The references are proof that interlinked biases exist…” however those referred interlinked biases have been explored, described, drawn attention upon, in a broad range of bibliography, published since early 2020. Consequently, is yet important to include the author’s criticism of those references, as to highlights the theories, methods, techniques, that other related works have provided from 2020, that are useful in their pursue of the design of a new framework for “…mitigating bias from practical ML model pipelines.” The manuscript needs to explicitly showcase the elements, from the studied theories, that are considered useful, so as those considered the opposite (and why) for a further design of a new framework for Bias mitigation. Additionally, I once more suggest to include criticism of the available frameworks for Bias mitigation/reduction/avoidance, present in the available literature, that still have not been included in the manuscript. I consider this issue important, as is the one that will help the authors delineate at what extent their work distinguishes from the ones already available in the field; what their contribution is, aside of a recollection of works that proof Bias exist in ML and AI context, which others works are already successfully proved.
And secondly, the manuscript is still lacking of the analysis of an operationalization of the interlinked biases whose existence the authors are so focused on proving. At this regard the authors claim “…we find the working definitions for Bias, Fairness and other variables in the context of AI and ML is premature at this point.”, failing to consider that those variables are the foundations of any further possible solution proposal that might outcome from their survey. My logic is the following, what working variables will be the ones that will support the design of the framework that the authors will end proposing as a consequence of this study? Without an understanding, or at least delineation/scoping of these variables, the next step in their research method will be, without room for doubt, another literature survey, this time, to explore and analyze the referred working variables. The manuscript, as is, like the authors have so assertively pointed in their rebuttal letter, only draws attention to the existence of interlinked Biases in a given context, like many other available literature. This issue is important when designing a framework that will be dedicated to mitigate/reduce Bias, as will provide a better, concept map –to put it somehow in plain text- of the variables that the variable Bias can be operationalized in.

Experimental design

Closely related, and in complete coherence with the previous issues described in the section "Basic Reporting", I would like to insist in the inclusion of the analysis of agile methodologies for AI and ML solutions, particularly the ones within the environment of the scope of the manuscript. This issue is important as it provides insights on the parts of the developing cycle the Bias is mostly introduced, and how that might be tackled once the proposal framework (the authors claim they will further design) is introduced.

Validity of the findings

I have no additions to my previous review in this section.

Additional comments

The issues described in the sections "Basic reporting" and "Study design" that are still unsolved, in my opinion, should be highly considered before the final acceptance of the manuscript.

The authors must reflect on the purpose of their literature survey. Their paper – once published - can, and will, if done thoughtfully, constitute not only a cornerstone for their future works, but a highly valuable tool for many other researchers and AI/ML developers and practitioners included in their target audience.

---

## Round 0.3 · accepted · Accept

Dear Authors, The reviewer is now satisfied with your updates. Thus I have decided to accept your manuscript. Regards,

·

Basic reporting

The authors have addressed my suggestions in this regard or have successfully explained their strategy in their rebuttal letter.

Experimental design

The authors have addressed my suggestions in this regard or have successfully explained their strategy in their rebuttal letter.

Validity of the findings

The authors have addressed my suggestions in this regard or have successfully explained their strategy in their rebuttal letter.

Additional comments

The authors have addressed my suggestions in this regard or have successfully explained their strategy in their rebuttal letter.